# Examining interviewer bias in medical school admissions: The interplay between applicant and interviewer gender and its effects on interview outcomes

**Stefanos A. Tsikas**[ORCID]*, **Karina Dauer**

Dean of Studies Office, Academic Controlling, Hannover Medical School, Hannover, Lower Saxony, Germany

* Tsikas.Stefanos@mh-hannover.de

## Abstract

Selection interviews have long been integral to medical school admissions, yet their limited predictive validity and susceptibility to bias raise concerns. This study delves into potential interviewer bias within the dynamics of interviewee and interviewer gender. We analyze a dataset of 5,200 applicants and over 370 selection committees engaged in semi-structured interviews from 2006 to 2019 at a large German medical school with multiple linear and non-linear regression analyses. Our findings reveal that all-female committees tended to award male candidates, on average, one point more than their female counterparts, significantly enhancing the chances of submission for male applicants despite lower academic grades, which constituted 51% of the selection process points. All-male and mixed-gender committees exhibited similar ratings for both genders. The role of valuing voluntary services emerged prominently: all-male and mixed committees acknowledged women's volunteer work but not men's, while all-female committees demonstrated the opposite pattern. Our results attribute variations in interview outcomes to the absence of standardization, such as insufficient interviewer training, divergent rating strategies, variations in interviewer experience, and imbalances in candidate allocation to selection committees, rather than to a "gender bias", for example by favoritism of males *because* of their gender.

## Introduction

### Background

Interviews have been widely used as a selection method for medical schools and residency programs, serving as a platform to assess candidates' personal requirements, motivation, social and communication skills. They can provide valuable information on non-cognitive soft skills that are often overlooked in other, predominantly cognitive, selection criteria [1–4]. However, the use and value of interviews as a selection criterion is debatable, as they have been found to be unreliable predictors of later study success, compared to cognitive criteria such as high school GPAs or aptitude tests [1, 2, 5–9].

**Funding:** The author(s) received no specific funding for this work.

**Competing interests:** The authors have declared that no competing interests exist.

There is some evidence that interview outcomes correlate with personality-oriented psychometric tests [10] and that interview assessments may predict later performance-evaluations that are also on a practical-personal level [11]. Nevertheless, classical interviews are heavily context-specific and subjective [2, 10], and therefore prone to potential biases that may affect interview outcomes and student selection.

Previous research on potential biases in selection and residency interviews at medical school, or in patient evaluations of physicians, addressed various factors, such as gender, nationality/ethnicity [12–15], age [16], socioeconomic background [14, 17, 18] or personality type [19–21]. Empirical evidence for biases is mixed, and if present, they can be rooted in both the characteristics of interviewers and interviewees, potentially harming the reliability as well as the validity of interviews [22].

In this study, we focus on the impact of gender on interview outcomes in semi-structured selection interviews conducted at a German medical school over a period of 14 years, covering 5,200 successful and unsuccessful candidates and over 370 interview committees. Our objective is to investigate and explain potential interviewer biases, and to provide some insights into how biases can be mitigated or prevented to ensure fair and objective selection processes.

To contextualize our research, we provide a brief review of related literature on gender-bases interviewer biases in selection interviews below. At the end of this introductory section, we describe our research objectives in more detail.

## Gender bias in selection interviews

Gender bias, and specifically bias against women, has been extensively researched in many settings and in various academic disciplines. Specifically in job-interviews, Isaac et al. [23] showed with a systematic review that gender bias against women (lower ratings than equivalently competent men) is often present, in particular in "male sex-typed jobs" (p. 1440).

Relatively little research has been done on interviewer biases in selection interviews for higher education programs, and specifically medical studies–in our paper, we aim to address this research gap.

Marquart et al. [24] found that the content of interviewer questions in unstructured selection interviews varied significantly with applicants' gender (women were more often asked personal questions than men), and that interviewees were more comfortable talking to interviewers with the same gender. Griffin and Wilson [25] found that female interviewers at an Australian medical school rated both, male and female candidates, with higher overall scores in a Multiple-Mini Interview, compared to male interviewers. The authors suggest that this rating behavior might be affected by gender-specific differences in personality. In a contrasting finding, Wilkinson et al. [26] found that gender disparity between females and males increased in favor of the latter after a multifaceted admission test (called GAMSAT) was no longer supplemented by selection interviews at an Australian medical school, because males scored higher in the test. The authors went as far as to argue that interviews could be an important method to ensure a fair gender proportion in medical programs. Analyzing a single entry year to a large UK medical school, another study found that interview scores were not affected by gender, ethnicity or socio-economic background of interviewees and interviewers [27]. For the admission process at Brown University, Smith [28] showed that eliminating personal interviews did not alter medical students' characteristics, e.g. regarding gender, ethnicity, or academic performances. In recent studies on interviews for residency training, no biases were identified [29, 30].

Because of the described shortcomings of personal interviews, structured and standardized formats have become more and more popular. A frequently used instrument is the Multiple

Mini-Interview (MMI), which mimics the structure of Objective Standardized Clinical Examinations (OSCE). MMIs have found to be less dependent on interviewer characteristics, less prone to biases [16, 31, 32], and they have higher predictive validity for study success compared to other interview types [33–36].

### Research objectives

In this article, we explore whether interview ratings and the selection of students in a semi-structured interview was influenced by and mutually dependent on the gender of both the interviewees and the interviewers.

We aim to introduce several innovations and expansions compared to previous research on this topic: We analyze not only individual years, but rather all interviews conducted over a span of 15 years at a large German medical school. This provides a significantly larger dataset and greater representativeness of the examined selection process. In our study, we focus on interaction effects between the gender of interviewees and interviewers and thus aim to go beyond simple group-comparisons. Instead of merely reporting differences, we aim to provide explanatory approaches. For instance, we examine whether certain characteristics, such as the presence of voluntary service, can account for differences and whether these are gender-specific. The structure of the selection process allows us to incorporate high school graduation grades as a cognitive selection criterion and to investigate their role in the interview process. Specifically, we explore whether interviews had a superior or subordinate role for student selection based on the gender of applicants. This is particularly relevant as women, due to better academic performance in school, had a "head start" in the selection process.

## Materials and methods

### Student selection procedure

Each year approximately 400 applicants, who had been preselected based on their high school grades and had listed the Hannover Medical School (MHH) as their first preference for studying medicine, were invited to the selection interviews. Only applicants with good to very good grades (thus, an A or at least B+) had the chance to participate. The interviews aimed to assess social competence, societal engagement, motivation to study, and self-reflection–in other words, those qualities that are only inadequately reflected in high school grades or aptitude tests.

Student selection at MHH was based on a ranking of a weighted 'Total score' in the range of 0 (worst) to 15 (best) points, which consisted of 49% 'Interview grade' and 51% 'Abitur grade'. The latter was included in the selection procedure because of laws stipulating that the high school (or Abitur) grade had to be the dominant selection criterion. Zero points were equivalent to the high school leaving grade 4.0 ('D'), the worst passing grade. The maximum 15 points corresponded to the best Abitur 1.0 ('A+'). Interviews were supposed to last at least 20 minutes but no longer than 30 minutes.

The conversation between candidates and selection committees was designed to be relatively open and, as sketched in the left picture of Fig 1, focused on three pre-defined conversation topics. Under personal qualifications (Q-Pers), committees evaluated how candidates presented themselves regarding empathy, responsiveness, linguistic expression, and openness, as well as whether a reflective engagement with personal strengths, weaknesses and ideals was apparent. Under professional aspects (Q-Pro), committees evaluated the motives for studying medicine at MHH, as well as the motivation and reasoning for choosing academic subjects in school. Interviewers also asked about knowledge of the content and structure of medical studies, as well as concrete ideas about the medical profession. Under extracurricular activities

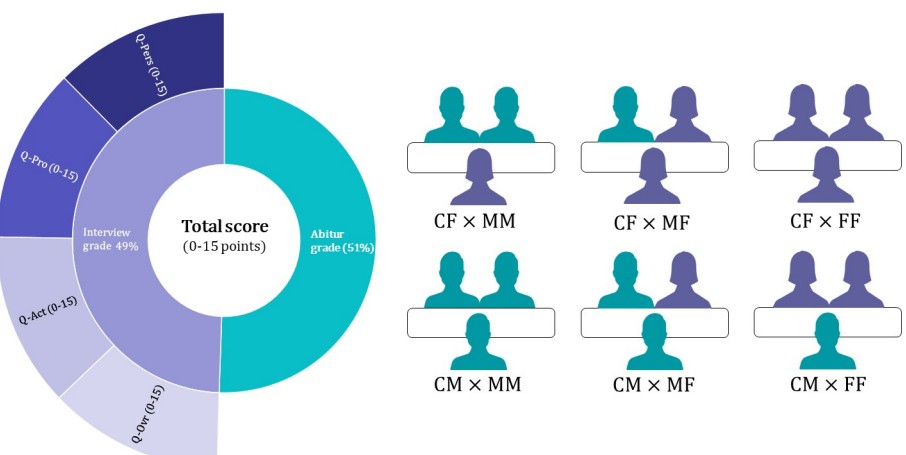

**Fig 1. The student selection procedure at Hannover Medical School (MHH).** Left picture: Admission to medical school was based on a ranking of participants' 'Total score'. The total score was calculated by 0.49 × Interview grade + 0.51 × Abitur grade, both a score running from 0 (worst) to 15 points. Right picture: a female (CF) or male (CM) candidate was interviewed by either two males (MM), two females (FF) or a mixed selection committee (MF), resulting in six interviewer-interviewee pairs.

(Q-Act), the committees evaluated the candidates' engagement with literary and scientific topics, musical interests, volunteer and other activities. In addition, a purely subjective overall impression (Q-Ovr) was rated, for which no predefined criteria were used.

Prior to the interviews, all invited applicants were required to answer a short questionnaire that provided interviewers information on e.g. school subjects, hobbies, vocational training or experiences as foreign exchange students.

The interviews were not standardized, meaning that no uniform questions were asked to all candidates. The content of the interviews likely varied widely between committees, and even within individual committees, there was no prescribed recurring thread of conversation. Prior to the selection interviews, interviewer trainings and general agreements were rather cursory, and participation was de facto not mandatory.

Each selection committee consisted of two physicians who had to be faculty members at MHH. Committees were formed in a way that members belonged to different sections (i.e. large and small clinical subjects, theoretical institutes), and that at least one interviewer was a clinically active doctor. As outlined in the right picture in Fig 1, this resulted in six possible interviewer-interviewee pairs: a male candidate (CM) or a female candidate (CF) could be interviewed by either an all-male (MM), mixed-gender (MF), or all-female (FF) committee, resulting in the 2×3 matrix shown in Fig 1 (right picture). Prior to the interviews, participants were randomly assigned to committees.

The commissions evaluated each of the four interview categories on a scale from 0 (worst) to 15 points, which determined 25% each of the overall interview grade. After the conversation, the two commission members agreed on a common score for each category. Therefore, we can only investigate potential interviewer bias at the commission level and not at the individual level.

It is important to note that the commissions did not directly determine who was selected and who was not, as only the interview grade was transmitted. For lower scores, the interviewers could be certain that a candidate would not receive a study place. Conversely, a very high interview grade did not necessarily guarantee acceptance.

## Data

In this study, we use data from 5,200 individuals who participated in the selection interviews between 2006 and 2019 (interviews have not been conducted since 2020). On average 370 interviews per year are included in our data, around 150 applicants were admitted annually. Not all invited candidates participated, and we exclude some observations where it was, due to incomplete information, not certain whether an interview grade of 0 was due to a very poor performance or whether the candidates did not attend the interview but were still included in the final ranking.

For each interviewee, we have core socio-demographic information as listed in Table 1 below, and know the gender composition of the respective selection committee, thus, if two males, two females, or a mixed pair interviewed a candidate (see Table 2 for interviewer statistics).

**Table 1. Descriptive sample statistics.**

| Variable | Not selected | Selected | Overall |
|---|---|---|---|
| Abitur grade | 12.48 | 12.91* | 12.65 |
| (SD; Median) | (1.11; 12.5) | (1.08; 13) | (1.12; 13) |
| Interview grade | 9.33 | 14.02* | 11.27 |
| (SD; Median) | (2.82; 10) | (0.96; 14) | (3.22; 12) |
| Overall grade | 10.87 | 13.45* | 11.93 |
| (SD; Median) | (1.57; 11.15) | (0.68; 13.5) | (1.81; 12.26) |
| Age at interview date | 19.91 | 19.78* | 19.86 |
| (SD; Median) | (2.38; 19) | (19.78; 1.90) | (19.86; 2.20) |
| Waiting semester | 1.12 | 1.07 | 1.10 |
| (SD; Median) | (1.97; 0) | (1.93; 0) | (1.96; 0) |
| Females (%) | 70.66 | 67.22* | 69.27 |
| Voluntary service (%) | 14.26 | 16.12 | 15.01 |
| Abitur Gymnasium (%) | 83.42 | 84.93 | 84.05 |

Mean values, standard deviations and median values are in parentheses.

*: Statistically significant difference ($t$-test; $p<0.05$) between selected and not selected participants.

**Table 2. Selection committee statistics.**

| Variable | MM | MF | FF |
|---|---|---|---|
| % of selection committees | 55.88 | 40.91 | 3.20 |
| % of candidates interviewed | 58.18 | 39.19 | 2.64 |
| % of candidates selected | 40.78 | 39.95 | 40.45 |
| Female candidates (%) | 68.25 | 70.48 | 71.32 |
| Abitur grade | 12.60 | 12.74 | 12.56 |
| | (1.11;12.5) | (1.17;13) | (1.21;12.5) |
| Interview grade | 11.26 | 11.27 | 11.36 |
| | (3.23;12) | (3.23;12) | (3.06;12) |
| Overall grade | 11.88 | 11.99 | 11.93 |
| | (1.81;12.22) | (1.81;12.32) | (1.81;12.26) |

Mean values, standard deviations and median values are in parentheses. MM: all-male selection committee; MF: mixed-gender committee; FF: all-female selection committee.

For some years, data on ratings in the separate interview segments, candidates' age and educational background (type of school and federal state in which the Abitur was completed) are missing. It is worth noting that 98% of our sample are German citizens, also because international students are primarily admitted upfront via a preliminary quota. In the course of this paper, we define observed differences in the data as statistically significant when $p < 0.05$ and set 95% confidence intervals accordingly.

## Ethics statement

The present study analyzes only retrospective, administrative data on educational outcomes and does not constitute research with human subjects in a clinical sense. Only fully anonymized data was analyzed. All applicants at Hannover Medical School (MHH) gave consent, by agreeing to the MHH's study regulations (sect. 14, para. 1–5, 'Immatrikulationsordnung'), that personal data may be used for evaluation/research and quality assurance purposes. This is in accordance with the Higher Education Act (in particular sect. 17, para. 3 *NHG* and sect. 3, para. 1, cl. 1 *NHG*) in Lower Saxony, Germany. The study regulations have passed all governing bodies at MHH, rendering a separate approval by an ethics committee unnecessary. The research presented here is in accordance with the Helsinki Declaration. Data on interviews and selection committees were accessed on February 14, 2023. After data collection, a fully anonymized dataset was generated for data analysis.

## Empirical approach

Our empirical approach centers on utilizing the factorial design of interviewer-interviewee pairs illustrated in Fig 1. We aim to explore whether interview ratings and admission probabilities differ between men (CM) and women (CF) and whether these differences are influenced by the three committee types (MM, MF, FF). We refrain from employing simplistic significance tests for group differences, as we assume that factors beyond the predictors of interest candidate gender and committee composition affect the outcomes. For instance, over the course of 14 years of the selection process, high school grades significantly increased ($r = 0.67$). Beyond this, a better high school grade could potentially correlate with both interview performance and influence interviewer assessments. Furthermore, voluntary service (as an additional information about candidates) might affect interview ratings, as it signals commitment, dedication, and social skills–qualities that interviewers should explicitly appreciate. Only 15% of all candidates had completed voluntary service, with an imbalance between CM (22%) and CF (11%).

To account for these influencing factors, we estimate an OLS regression with the interview points of candidate $i$ (overall and in the four sub-categories) as the dependent variable. The 2×3 design of interviewer-interviewee pairs is incorporated into the linear regression model as an interaction term $commission_j \times female_i$, where $j \in [0,1,2]$ (0: = MM, 1: = MF, 2: = FF) and *female* takes the value 1 when candidate $i$ = CF. As interaction effects can be challenging to interpret in tables, we calculate partial effects and depict them graphically (Fig 2). Year fixed effects ($FE_t$) are included to control for time-dependent trends or year-specific circumstances, such as increases in high school grades over time or the distribution of committee types.

As previously described, the 'Total score' determining admission results from both interview and high school grades. To investigate how these components (in mutual dependency) influence the admission probability, we estimate a probit regression in the 'Selection probability' subsection of the Results chapter, with the binary indicator *Selection* (yes = 1, otherwise 0) as the dependent variable. This approach allows us to make statements about whether the admission probability for men and women depended on the selection committee type and

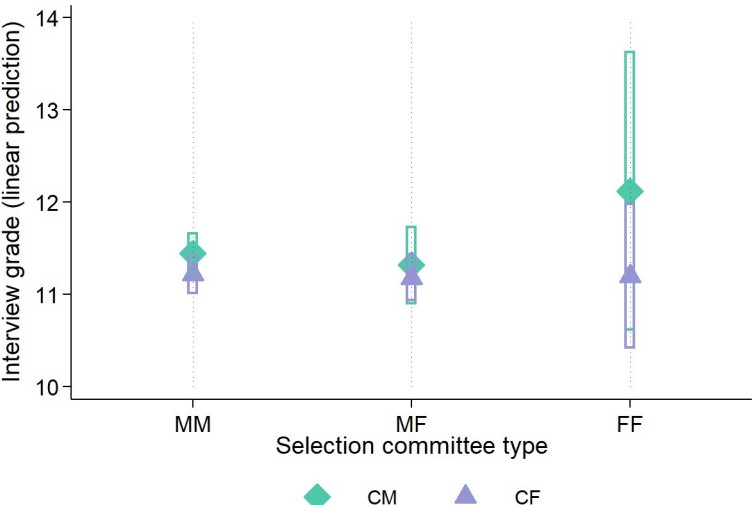

**Fig 2. Predicted interview grades for CM & CF, by composition of selection committees.** Fig 2 is the graphical representations of partial effects based on the interaction term in column (1) of Table 3. See the S1 File for further information. 95% confidence intervals are indicated. CM: male candidate; CF: female candidate; MM: all-male selection committee; MF: mixed-gender selection committee; FF: all-female committee.

whether MM, MF, and FF interacted differently in the handling of interview grade and high school grade. In the probit model, we control for the same covariates as in the linear regressions. Age and educational background are only included in robustness and sensitivity checks detailed in the S1 File, as these variables are missing for certain years. The results of the probit regressions for admission probability are also graphically represented, with a full documentation in tables provided in the S1 File.

## Results

### Descriptive sample statistics

In Table 1 we report basic sample statistics, which we differentiate by selected and not selected candidates to give an idea which parameters were decisive in the student selection process at MHH. Overall, 40.5% of interviewed candidates were admitted to study, with male candidates (CM) having a higher admission rate at 43.2% compared to female candidates (CF) at 39.3% (this difference is statistically significant, *t*-test). These ratios have remained stable over time. Overall, 69% of interviewees were female (selected: 67%; not selected: 70.7%). Although successful candidates have had a significantly better Abitur grade than unsuccessful candidates, the decisive criterion was the interview [37]. In the 'selected' sub-group, the standard deviation is very narrow. This suggests that only participants with an excellent interview performance had the chance of being admitted, and many persons were not selected despite having a very good total score. Tsikas & Fischer [37] have shown that the interview performance was the determining criterion for student selection at MHH despite the larger weight assigned to the Abitur.

In our sample, selected persons were slightly younger than the unsuccessful. In Table 1, we do not find statistically significant differences with respect to completed civil service, waiting time, and whether the Abitur was completed at a regular 'Gymnasium' (i.e. an academic high school) or at other schools, e.g. integrated schools or night schools. While 22% of male

candidates had completed a voluntary service, the share among females is 11%. In FF committees, the ratio is 3:1.

Over the 14 years of interviews, 56% of selection committees were comprised of two males (MM), who interviewed 57% of all participants (Table 2). MF groups accounted for 41% of committees, interviewing 40% of candidates, while all-female committees (FF) constituted 3% of groups and interviewed circa 3% of participants. Over time, two noteworthy changes occurred: the number of selection committees increased steadily from 16–20 in the early years to more than 30 towards the end, and the share of MF groups also increased, constituting the majority of committees in recent years. The maximum share of participants interviewed by FF committees was 7.2% in 2017, and in five years, no FF groups conducted interviews.

Table 2 further shows that the overall share of admitted students is relatively consistent across all committee types, as is the share of female interviewees. We also do not find statistically significant differences (*t*-tests) in interview grades between committee types. However, we do find significant differences in Abitur grades, though they are relatively small: candidates interviewed by MF had better Abitur grades than MM and FF (the difference between MM and FF is also statistically significant).

## Interview ratings

Table 3 presents the results of linear regression analyses, with the overall interview score (column (1)) and the scores in specific conversational categories (columns (2)-(5)) as the dependent variables. The predictors of interest are the gender of the participants (CF, CM) and the composition of the selection committees (MM, MF, FF).

The findings from Table 3 suggest that there are no statistically significant differences in interview ratings. In general, FF committees awarded candidates approximately 0.67 more points in the interview compared to MM committees. The negative coefficient sign of the FF × CF interaction term implies that this was due to better ratings for CM relative to female candidates. For the interview categories Q-Pers and Q-Pro, the estimated differences in FF ratings of CF and CM are even greater, although not statistically significant. Across all specifications in Table 3, differences in coefficient magnitude are smaller when comparing committee types MM and MF.

Across all interview subcategories, a stronger Abitur grade is associated with a higher interview score, though the correlation is only moderate. Interestingly, completing voluntary service is significantly linked to an increase of approximately one additional point in the interview across all specifications in Table 3, and not limited to Q-Act, where such experiences would be an evident topic of conversation. Drawing meaningful conclusions from interaction terms within a regression table can be challenging. To enhance interpretability, we calculate the partial effects of regression model (1) with respect to the committee type. This yields the predicted interview score of CF and CM for each committee type, adjusted for the covariates included in the regression, as depicted in Fig 2.

Fig 2 corroborates that there are no significant differences between CF and CM for any committee type. However, FF committees scored males approximately one point higher than females, while the differences in MM and MF groups are marginal. FF committees rated CF similarly to MM and MF. As the total number of interviews conducted by all-female interviewers is limited (as seen in Table 2), the 95% confidence intervals (represented as bars in Fig 2) are notably wide.

The preceding results prompt the question of whether the appreciation of voluntary service was gender-specific and/or influenced by the varying committee types. Fig 3 provides a visual representation of a regression model, wherein we introduced an interaction between the

**Table 3. Regression analysis for interview scores.**

| Dep. Variable | (1) | (2) | (3) | (4) | (5) |
|---|---|---|---|---|---|
| | **Interview grade** | **Q-Pers** | **Q-Pro** | **Q-Act** | **Q-Ovr** |
| **Selection committee type (reference is MM)** | | | | | |
| MF | -0.132 | -0.027 | -0.049 | 0.128 | 0.003 |
| | (0.177) | (0.195) | (0.201) | (0.197) | (0.198) |
| FF | 0.672 | 0.778 | 0.849 | 0.322 | 0.608 |
| | (0.516) | (0.565) | (0.535) | (0.601) | (0.563) |
| **Gender of applicant (reference is CM)** | | | | | |
| CF | -0.234 | -0.089 | -0.320* | -0.317* | -0.258 |
| | (0.129) | (0.143) | (0.149) | (0.148) | (0.148) |
| **Interaction terms (reference is MM × CM)** | | | | | |
| MF × CF | 0.097 | 0.119 | 0.217 | 0.137 | 0.180 |
| | (0.206) | (0.225) | (0.234) | (0.228) | (0.229) |
| FF × CF | -0.685 | -0.907 | -0.707 | -0.016 | -0.723 |
| | (0.596) | (0.641) | (0.651) | (0.680) | (0.654) |
| Abitur grade | 0.187* | 0.204* | 0.207* | 0.148* | 0.193* |
| | (0.061) | (0.068) | (0.073) | (0.071) | (0.071) |
| Voluntary service | 0.878* | 0.932* | 1.158* | 1.050* | 1.077* |
| | (0.140) | (0.150) | (0.152) | (0.152) | (0.153) |
| Waiting time | -0.029 | -0.068* | -0.018 | -0.070* | -0.055 |
| | (0.033) | (0.034) | (0.036) | (0.036) | (0.036) |
| Year FE | YES | YES | YES | YES | YES |
| Observations | 5036 | 4371 | 4371 | 4371 | 4371 |
| $R^2$ | 0.021 | 0.021 | 0.028 | 0.025 | 0.023 |

OLS regressions for the overall interview grade and the separate conversation topics (see Fig 1). Robust standard errors are in parentheses. The number of observations in columns (2)-(5) is smaller because data is missing for two years. Year-fixed effects (FE) are jointly significant in all specifications (Wald-Test). See Table A1 in the S1 File for extended models including Age and School type as additional control variables. The underlying regression model is described in the S1 File (see equation (1)) as well. CM: candidate is male; CF: candidate is female; MM: all-male selection committee; MF: mixed-gender selection committee; FF: all-female committee.
*: $p < 0.05$

voluntary service indicator and the committee type-interviewee gender interaction, as utilized in Table 3.

Panel (a) in Fig 3 illustrates the predicted interview score for committee types when the interviewee was female. Across MM, MF, and FF committees, female candidates without voluntary service received approximately 11 points. In MM and MF committees, female candidates with voluntary service achieved slightly above 12 points, a statistically significant difference. Within FF groups, this difference is nearly nonexistent. However, the confidence interval for females with voluntary service is quite broad due to a limited number of interviews, especially with females who had completed voluntary service.

In panel (b) of Fig 3, we observe a different trend for male candidates. Although MM and MF committees assigned a slightly higher rating to CM with a service, the disparity is minimal and statistically insignificant. In contrast, FF committees granted males with a voluntary service about one point more than CM without service and almost 1.5 points more than females with voluntary service. The wide confidence intervals, however, render this finding statistically insignificant. Similar to Fig 2, the results are adjusted for Abitur grade, waiting time, and interview year.

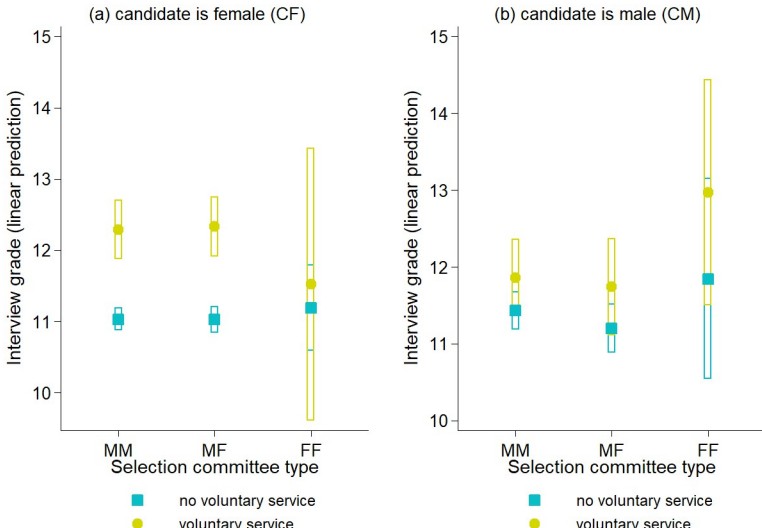

**Fig 3. Predicted interview grade of female (a) and male (b) candidates by status of voluntary service.** The plots are based on equation (2) discussed in the S1 File. 95% confidence intervals are indicated. MM: all-male selection committee; MF: mixed-gender selection committee; FF: all-female committee.

## Selection probability

While FF committees did not award significantly more points to men in the interview than to women (Fig 2), Table 4 reveals that the difference of more than one point significantly influenced the ranking of candidates. With approximately 370 interviews conducted annually, men in FF committees achieved an average rank position (RI) of 146. This is 43 places lower and thus statistically significantly better (indicated by the symbol †) than that of female candidates. Men also obtained a significantly better mean interview rank in MM and MF groups, with a difference of approximately 12 places. The average interview rank of CF is comparable in all committee types.

Of course, the selection interview contributed to just under half of the final admission decision; the second factor was the high school grade. Table 4 also displays the average position in

**Table 4. Comparisons of average rankings in Abitur (RA) and interview grade (RI).**

|  | MM | | MF | | FF | |
|---|---|---|---|---|---|---|
|  | **Mean RI** | **Mean RA** | **Mean RI** | **Mean RA** | **Mean RI** | **Mean RA** |
| CM | 172.23* | 188.38 | 177.71* | 192.58 | 145.95* | 216.05 |
|  | (927; 108.8) | (927; 110.02) | (584; 108.52) | (584; 108.46) | (39; 111.06) | (39; 90.18) |
| CF | 185.84† | 183.55 | 188.86† | 189.00 | 189.08† | 172.74† |
|  | (2,006; 105.13) | (2,006; 107.35) | (1,385; 103.10) | (1,385; 107.79) | (96; 103.23) | (96; 104.18) |

RI is the ranking of participants' interview grade per interview year. Equivalent interview grades were randomly assigned with running tiers. RA is the ranking of participants' Abitur grade per interview year. Similar Abitur grades were again randomly assigned with running tiers. The average rank per participant gender and committee type is reported in the table. Only the ranking list of total scores (see Fig 1) determined acceptance or rejection of interviewees. Number of observations and the standard deviation of mean ranks are in parentheses (N; SD). CM: male candidate, CF: female candidate. MM: all-male selection committee, MF: mixed-gender selection committee, FF: all-female selection committee.

*: $p<0.05$ in a comparison of RA and RI within selection committee type and candidates' gender (*t*-test).

†: $p<0.05$ in a comparison of CM and CF for RA and RI within selection committee types.

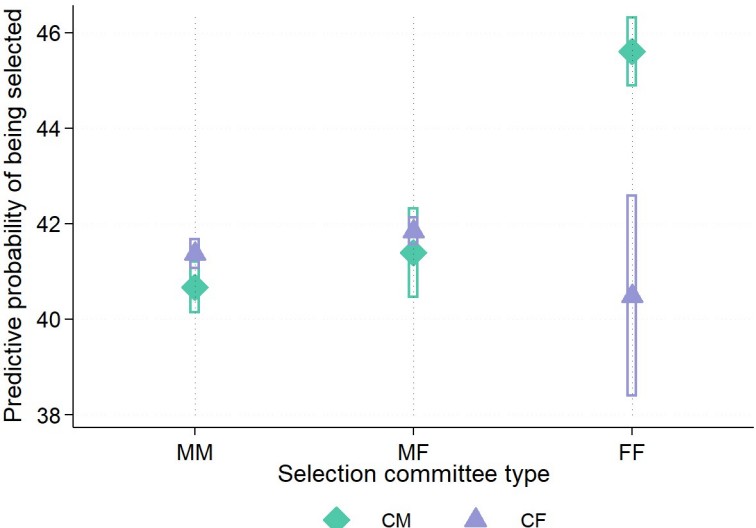

**Fig 4. Predictive probabilities for male/female candidates of being selected after the interview, plot by composition of selection committees.** Fig 4 is based on the regression model in column (1), Table A2 in the S1 File. 95%-confidence intervals are indicated. CM: male candidate; CF: female candidate; MM: all-male selection committee; MF: mixed-gender selection committee; FF: all-female committee.

the rank order of all high school grades (RA). While RA does not differ between men and women in MM and MF groups, CF in all-female selection committees had a significantly better high school grade than CM. In the comparison between committee types, the high school grade of CM in FF committees was below average, whereas that of CF was slightly better. In all three committee types, the average interview rank of men was significantly better than their selection rank (indicated by *), with the difference in FF groups being approximately 60 rank positions, much higher than MM and MF (approximately 15). The average interview assessment of CF in all committee types roughly corresponded to the high school grade.

The final selection decision was determined by the ranking list formed from the weighted points for the interview and Abitur grade. The estimated probability of admission (i.e., a binary yes/no decision) is depicted for all six possible interviewee-interviewer pairs in Fig 4. Similar to the regressions for the interview grade, the results in Fig 4 are adjusted for the same covariates that are included in Table 3.

The probabilities of admission for CM and CF do not differ significantly in commission types MM and MF. When two female interviewers conducted the interview, CM (45.6%) had a significantly higher probability of admission than CF (40.6%). Across commission types, the probabilities of admission for CF in Fig 4 do not differ significantly (the 95%-confidence intervals overlap). In FF groups, male candidates had a significantly higher probability of admission than in MM and MF.

It is important to reiterate that candidates brought their Abitur grade as a fixed value into the interview, and interviewers only influenced the chances of admission through the interview scores. Additionally, it cannot be ruled out that interviewers had information or inquired about the high school grades. This relationship between the candidates' gender, committee composition, and the components of the selection process is depicted in Fig 5, where the predictive probability of gaining admission (see Table A2 in the S1 File for a full regression output) is calculated for each level of Abitur grades.

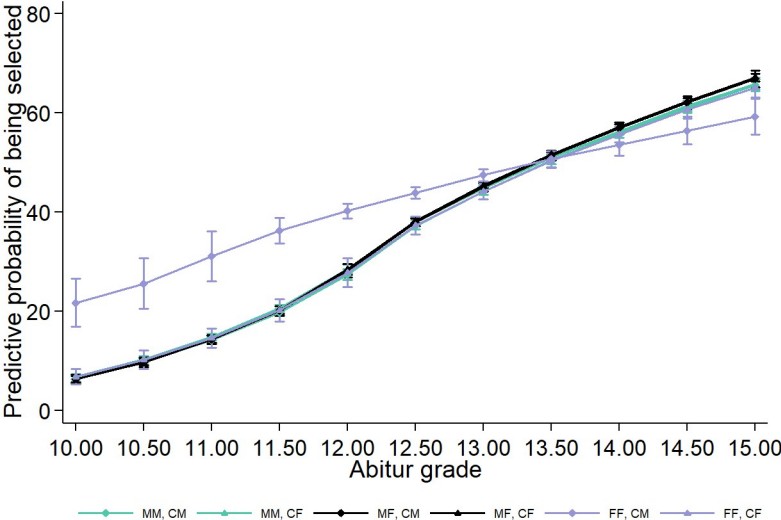

**Fig 5. Predictive probabilities for male/female candidates of being selected after the interview, dependent on the Abitur grade.** Fig 5 is based on the regression equation (3) in the S1 File and on regression model in column (3) of Table A2 in the S1 File. 95%-confidence intervals are indicated. CM: male candidate; CF: female candidate; MM: all-male selection committee; MF: mixed-gender selection committee; FF: all-female committee.

For five of the six interviewer-interviewee pairs, we find nearly identical results: for below-average grades, the probability of admission is below 10%, but it increases significantly with better academic performance. Only in the case of male candidates in FF committees is a different relationship observed. Even with lower grades, the probability of admission is around 20%, and it is significantly higher for all Abitur grade levels below 13 points compared to the other interviewer-interviewee pairs. From the previous analyses, we know that FF committees rated male candidates exceptionally well during interviews, and these male candidates simultaneously entered the interview with below-average academic performance compared to candidates in MM and MF groups.

## Discussion

### Summary of results and interpretation

In our study, we conducted an analysis of a comprehensive dataset comprising over 5,200 subjects and more than 370 selection committees involved in selection interviews at a major German medical school from 2006 to 2019. Beyond investigating gender-related differences in admission chances and interview ratings, we delved into various explanations both supporting and refuting the presence of gender bias.

The starting point for our analyses was the question of whether men and women were treated differently in the selection process, which combined interviews and high school grades, and if any potential differential treatment depended on the composition of the selection committees (MM, MF, FF). The answer to whether the interview rating and admission chances of men and women were influenced by the composition of the selection committees can be answered affirmatively, yet with noteworthy constraints:

All-female committees rated men in interviews more than a point higher than women, while differences in MM and MF committees were minimal. This rating discrepancy resulted in men being significantly better placed in an interview points ranking on average than

women, resulting in a significantly higher chance of admission despite weaker high school grades. However, the higher chances for men in FF did not disadvantage women, as there were no differences in interview points or admission chances compared to women in MM and MF.

Ultimately, it remains inconclusive whether higher ratings for men were due to gender, as reasons for the ratings and the course of the interviews were not documented in an analyzable form. In an influential experiment, Goldberg [38] found that women readers rated a male author ("John T. McKay") more favorable than a female author ("Joan T. McKay") of an identical article, and concluded that women had prejudices against their own gender. However, as Swim et al. [39] showed with a meta-study, Goldberg's result could often not be replicated, and overall gaps in ratings were negligible. Our data also suggests explanations counteracting Goldberg's claims [38]. For instance, candidates in FF committees differed slightly demographically from MM and MF (e.g., high school grades, voluntary services). From a procedural perspective, this may be considered potentially problematic if one assumes that the composition can influence evaluations and behavior. Legal requirements prioritized high school grades over interviews (a 51:49 split and pre-selection based on high school grades). Girls, on average, achieve better academic performance than boys, resulting in an approximately 65:35 distribution favoring women in selection processes that heavily rely on this criterion. A study by Tsikas & Fischer [37] at MHH suggests an informal tendency in the selection process to counterbalance high school grades with the interview. Candidates with excellent high school grades who failed to impress in the interview were rated so low that admission was unlikely. Conversely, compelling candidates in the interview were highly rated to potentially offset the disadvantage of weaker academic performance. This in particular could explain the behavior of FF committees, as candidates (predominantly CM) with below-average high school grades were over-represented here.

Our findings also suggest that extracurricular experiences such as a voluntary service have a positive impact on interview ratings. Interestingly, there is a gender-based disparity: MM and MF rated women with service significantly better than men. In FF, the opposite trend was observed. A voluntary service generally signals e.g. commitment, social skills, responsibility and dedication [40, 41] and can be beneficial on the job market [42, 43]. In addition, a voluntary service delayed applications for at least one year. In this time, candidates will have made valuable experiences and may have matured, which can positively affect self-awareness and self-presentation in the interview. Yet, this does not explain why such qualities were valued only in the gender opposite to the committee composition. After controlling for high school grade, interview year and other factors, we cannot reject that this result was perhaps gender-based. Another limitation is the small overall number of participants with a voluntary service, especially in FF committees, who interviewed only 3% of subjects.

## Contextualization

In line with existing literature, we believe that the observed inequalities and imbalances could have been prevented with a higher degree of standardization. This includes a clearly defined interview guide with questions posed to all candidates, comprehensive interview training and shared rating standards [25], and fewer interviewers conducting a larger number of interviews in fixed commissions, as this would constrain the focus, rating strategies, and personal preferences. Harasym et al. [22] showed that experienced interviewers rated candidates' true performance more precisely than novices. Although the distribution of candidates among committee-types was random, we believe randomization based on certain characteristics could have helped avoiding both unequal distributions in committees and differential ratings.

Furthermore, some studies point out benefits from explicitly incorporating vocational training, previous (job or educational) experiences or extracurricular activities into student assessment for medical school admission [21, 44, 45]. This might be indeed useful, as there is evidence suggesting that such experiences also have predictive validity for study success [45–48]. At MHH, interviewers may or may not have asked about vocational training or previous academic experiences, but there was no common understanding among interviewers how this should affect ratings (see our finding for the handling of voluntary services).

Recent studies examining residency interviews suggest that factors such as the date, time of day, and number of interviews may influence raters' assessments [49, 50], while others have found no effect [51–53]. Although we do not believe that these potential influences differ according to interviewers' gender, this remains an open and interesting question for future research on selection processes.

## Limitations

Similar to the few existing studies addressing biases in university admissions interviews, our investigation is a one-site study. Consequently, the results may not be generalizable due to the specificity of the selection procedure. We acknowledge an imbalance in the distribution of commission types, as only few female-only pairings interviewed applicants over the years, and candidates with below-average Abitur grades were slightly over-represented in FF committees.

On a more general level, we point out that our study did not aim to draw clear conclusions about whether any differences represent a "true" bias based on favoritism or discrimination e.g. due to gender-specific stereotypes. This was hindered by the lack of information and documentation regarding the reasons behind given evaluations.

We also did not aim contribute to biases, specifically gender bias, on a conceptual or theoretical level. The focus of this study was on a positive, empirical examination of potential interviewer biases embedded in the selection of applicants into medical school.

## Conclusion

Interviews are one of the oldest and still frequently used tools for selecting medical students, despite being expensive, time-consuming, and lacking predictive validity for academic success, unlike school grades or cognitive aptitude tests. Qualities like soft skills or motivation, which interviews aim to assess, are difficult to quantify, both in exams and in later professional activities of doctors. Moreover, evaluating personal qualities is prone to biases. Interview biases can stem from procedural structures, especially with a lack of standardization. Biases may also arise when interviewers try to compensate for perceived disadvantages of certain groups. Based on our data, we conclude that both explanatory approaches could be conceivable in our study with men interviewed by all-female selection committees. If selection interviews are conducted, high standardization, careful interviewer training and diverse and distinct interview elements should be prioritized. Among known procedures, Multiple Mini-Interviews best meet these criteria.

## Supporting information

**S1 File. Supplementary information and tables.**
(PDF)

**S1 Data.**
(XLSX)

## Acknowledgments

We thank Stefanie Bögeholz for her help and assistance in the provision of data. We thank our colleagues Volker Paulmann and Volkhard Fischer for helpful comments and engaging discussions. We thank two anonymous reviewers for their helpful suggestions.

## Author Contributions

**Conceptualization:** Stefanos A. Tsikas, Karina Dauer.

**Data curation:** Stefanos A. Tsikas, Karina Dauer.

**Formal analysis:** Stefanos A. Tsikas.

**Investigation:** Stefanos A. Tsikas.

**Methodology:** Stefanos A. Tsikas.

**Software:** Stefanos A. Tsikas.

**Writing – original draft:** Stefanos A. Tsikas, Karina Dauer.

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
