## [Decision Letter · Decision Letter 0]

8 Jul 2024

PONE-D-23-41751Examining interviewer bias in medical school admissions: The interplay between applicant and interviewer gender and its effects on interview outcomesPLOS ONE

Dear Dr. Tsikas,

Thank you for submitting your manuscript to PLOS ONE. After careful consideration, we feel that it has merit but does not fully meet PLOS ONE’s publication criteria as it currently stands. Therefore, we invite you to submit a revised version of the manuscript that addresses the points raised during the review process.

This manuscript will provide a valuable contribution to the literature.

Minor revisions, as suggested by the two reviewers, are required. I believe this round of minor revisions will increase the quality of the publication.

I encourage the authors to address the comments provided by the two reviewers, and I thank you for publishing your research on OLOS One. 

We look forward to receiving your revised manuscript.

Kind regards,

Umberto Baresi, Ph.D.

Academic Editor

PLOS ONE

2. For studies reporting research involving human participants, PLOS ONE requires authors to confirm that this specific study was reviewed and approved by an institutional review board (ethics committee) before the study began. Please provide the specific name of the ethics committee/IRB that approved your study, or explain why you did not seek approval in this case.

4. In the online submission form, you indicated that [Data and code can be made available by the corresponding author upon reasonable request].

Reviewers' comments:

Reviewer's Responses to Questions

**Comments to the Author**

1. Is the manuscript technically sound, and do the data support the conclusions?

Reviewer #1: Yes

Reviewer #2: Yes

2. Has the statistical analysis been performed appropriately and rigorously? 

Reviewer #1: I Don't Know

Reviewer #2: Yes

3. Have the authors made all data underlying the findings in their manuscript fully available?

Reviewer #1: No

Reviewer #2: Yes

4. Is the manuscript presented in an intelligible fashion and written in standard English?

Reviewer #1: No

Reviewer #2: Yes

5. Review Comments to the Author

Reviewer #1: Thanks for the interesting data about a relevant topic. The reader need some important informations in the methods section and also the discussion would benefit from some more thoughs.

Language

I am not a native speaker myself, but I had the impression, that the language could be improved. E.g. “Motivated by the described shortcomings of personal interviews…,” or: Or (page 15, line 4): ….”Missing for some years are data on ratings in the….” sounds very “German”. Same with “..One often-used instrument is the Multiple Mini-Interview….” Better: a frequently used instrument as the MMI….”. Please check the language once more through the whole paper.

Please correct also upper and lower case letters (e.g. in methods: “Results chapter”)

Abstract

Ok, if no subheadings are needed

Introduction

Page 10, 3 paragraph: delete the first word “in” or “has been found that….”

Same paragraph: what sort of “test” is meant here: “……in favor of the latter males scored higher in the test” ? interviews? MMI? Cognitive or psychometric tests?

Same paragraph: “For the admission process at Brown University, [28] showed that…” please add authors name or “it was shown, that….”.

Research objectives:

delete the sentence: “(as discussed in the previous section)”

Some sentences can moved into the discussion

e.g.: into “strength”: data from 15 years,

or into “limitations”: one-site study, no detailed information about rating

Methods

In the methods-section, several questions are not answered so far:

Did the interviewers know the candidate's A-level grades beforehand? Or other information about the candidates?

What was the training like when it took place?

Which professional groups did the interviewers come from?

How many minutes did each interview last?

What is the maximum number of consecutive interviews per day/per session? Please discuss later possible exhaustion after a long periods as it is known from OSCES or exams.

Were all candidates binary or were some divers?

Why data collection ended in 2019?

Were the intervievers informed about the candidates success later?

Results

good

Discussion

Please add subheadings.

P 20., last line: something missing? The reader expects a quotation („Goldberg claims…“).

How is he distribution between male and female students at MHH in the years of study? Perhaps female interviewers had a sort of “hidden agenda” to increase the proportion of male students?

Please add some ideas about further research

No conclusion

Literature

Ok

Graphs/tables

good

Reviewer #2: In this study “Examining interviewer bias in medical school admissions: The interplay between applicant and interviewer gender and its effects on interview outcomes” , the authors investigate gender-related interviewer bias in medical school admission interviews using data from 5,200 individuals who participated in selection interviews between 2006 and 2019.

I appreciate the time and effort spent by the authors on this important contribution to the literature. It is a study that will shed light on future research in this field. However, I suggest some changes before publication. These are minor technical issues that have come to my attention. The suggestions are listed below.

1. On Page 2, there is no need for "see reviews by" in the last sentence. Sources [1-4] are already mentioned. Please delete this statement.

2. In the third sentence on Page 3, there is no need for "reviews by" and "See also". Instead, list the sources side by side.

3. All of the following sources are written separately from each other. Please write them

[10] [2],

[14] [17-18]

[31-32] [16],

[44-45] [21].

4. On page 4, p. 1440, is this necessary? it is already referenced. This style of referencing is not customary for articles, nor is it necessary.

5. “For a start, we briefly present previous evidence on gender biases in (selection) interviews for study programs and residency trainings.”

I do not think these expressions are necessary, they even disrupt the flow. Going directly to the subject to be explained will help to understand more clearly.

6. It is sufficient to mention only the purpose of the research in the "Research objectives" section on page 5. Including the following section in the strengths and limitations section of the research will increase fluency. It should be moved to the relevant section

“Similar to the few existing studies addressing biases in university admissions interviews (as discussed in the previous section), our investigation is a one-site study. Consequently, the results may not be generalizable due to the specificity of the selection procedure. However, our study introduces several innovations and expansions compared to pre- vious research:

• Our data encompasses not only individual years but the analysis of all interviews con- ducted over a span of 15 years, providing a significantly larger dataset and greater rep- resentativeness of the examined selection process.

• The focus of the paper is on interaction effects between the gender of interviewees and interviewers and goes beyond simple group-comparisons

• Instead of merely reporting differences, we aim to provide explanatory approaches. For Instead of merely reporting differences, we aim to provide explanatory approaches. For instance, we examine whether certain characteristics, such as the presence of voluntary service, can account for differences and whether these are gender-specific.”

7. Likewise, the first paragraph on page 6 should be moved to the limitations section

“Some constraints of our study should be acknowledged upfront. Our study does not aim to draw clear conclusions about whether any differences represent a "true" bias based on favoritism or discrimination e.g. due to gender-specific stereotypes. This is hindered by the lack of infor- mation and documentation regarding the reasons behind given evaluations.

We do also not contribute to biases, specifically gender bias, on a conceptual or theoretical level. The focus is on a positive, empirical examination of potential interviewer biases embed- ded in the selection of applicants into medical school.”

8. Write MHH open under Figure 1

(Figure 1: The student selection procedure at MHH)

9. Page 21 --- A study by [37] ------ work done by whom? (please specify author)

10. Include the strengths and limitations of the study after the discussion section.

11. Finally, what is your take-home message? Please state it more clearly.

Good work

6. PLOS authors have the option to publish the peer review history of their article (what does this mean?). If published, this will include your full peer review and any attached files.

Reviewer #1: No

Reviewer #2: No

---

## [Author Response · Author response to Decision Letter 0]

1 Aug 2024

Our detailed responses to all reviewer and editor comments have been uploaded in a separate file.

---

## [Editor Report · Decision Letter 1]

9 Aug 2024

Examining interviewer bias in medical school admissions:

The interplay between applicant and interviewer gender and its effects on interview outcomes

PONE-D-23-41751R1

Dear Dr. Tsikas,

We’re pleased to inform you that your manuscript has been judged scientifically suitable for publication and will be formally accepted for publication once it meets all outstanding technical requirements.

Kind regards,

Umberto Baresi, Ph.D.

Academic Editor

PLOS ONE

Additional Editor Comments (optional):

We thank the authors for the thorough consideration of the reviewers' inputs.

We are pleased to communicate that the manuscript currently satisfies the requirements for publication in PLOS One.
---

## [Editor Report · Acceptance letter]

15 Aug 2024

PONE-D-23-41751R1 

PLOS ONE

Dear Dr. Tsikas, 

I'm pleased to inform you that your manuscript has been deemed suitable for publication in PLOS ONE. Congratulations! Your manuscript is now being handed over to our production team.

Kind regards, 

on behalf of

Dr. Umberto Baresi 

Academic Editor

PLOS ONE